# Worsening Symptoms Is Associated with Larger Cerebral Blood Flow Abnormalities during Tilt-Testing in Myalgic Encephalomyelitis/Chronic Fatigue Syndrome (ME/CFS)

**DOI:** 10.3390/medicina59122153

**Published:** 2023-12-12

**Authors:** C. (Linda) M. C. van Campen, Peter C. Rowe, Frans C. Visser

**Affiliations:** 1Stichting CardioZorg, Planetenweg 5, 2132 HN Hoofddorp, The Netherlands; fransvisser@stichtingcardiozorg.nl; 2Department of Pediatrics, School of Medicine, Johns Hopkins University, Baltimore, MD 21287, USA; prowe@jhmi.edu

**Keywords:** orthostatic intolerance, tilt-table testing, ME/CFS, stroke volume index, cardiac index, cerebral blood flow, disease severity, symptom worsening

## Abstract

*Background and Objectives*: During tilt testing, myalgic encephalomyelitis/chronic fatigue syndrome (ME/CFS) patients experience an abnormal reduction in cerebral blood flow (CBF). The relationship between this CBF reduction and symptom severity has not been examined in detail. Our hypothesis was that ME/CFS severity is related to the degree of the CBF reduction during tilt testing. *Materials and Methods*: First, from our database, we selected ME/CFS patients who had undergone assessments of ME/CFS symptomatology and tilt tests on the same day, one at the first visit and the second during a follow-up. The change in symptomatology was related to the change in CBF during the tilt test. Second, we combined the data of two previously published studies (n = 219), where disease severity as defined by the 2011 international consensus criteria (ICC) was available but not published. *Results*: 71 patients were retested because of worsening symptoms. The ICC disease severity distribution (mild-moderate-severe) changed from 51/45/4% at visit-1 to 1/72/27% at follow-up (*p* < 0.0001). The %CBF reduction changed from initially 19% to 31% at follow-up (*p* < 0.0001). Of 39 patients with stable disease, the severity distribution was similar at visit-1 (36/51/13%) and at follow-up (33/49/18%), *p* = ns. The %CBF reduction remained unchanged: both 24%, *p* = ns. The combined data of the two previously published studies showed that patients with mild, moderate, and severe disease had %CBF reductions of 25, 29, and 33%, respectively (*p* < 0.0001). *Conclusions*: Disease severity and %CBF reduction during tilt testing are highly associated in ME/CFS: a more severe disease is related to a larger %CBF reduction. The data suggest a causal relationship where a larger CBF reduction leads to worsening symptoms.

## 1. Introduction

Myalgic encephalomyelitis/chronic fatigue syndrome (ME/CFS) is a complex disease with dysregulation of multiple organ systems (including the central nervous system, immunological system, energy metabolism, the cardiovascular system, etc.), resulting in a large variety of symptoms [1,2]. To capture and assess the severity of these complaints, several questionnaires have been developed and applied, including the Bell CFIDS disability scale, the Chalder fatigue scale, the DePaul pediatric health screening questionnaire, the DePaul post-exertional malaise questionnaire, the DePaul symptom questionnaire, and many others [3,4,5,6,7]. However, these questionnaires are subjective and there is a risk of response bias. Therefore, these questionnaires should preferably be used in conjunction with objective measures of symptoms. Examples of these objective measures are the use of heart rate (HR) changes [8,9,10], heart rate variability (HRV) differences [9,11], activity trackers [12], and peak oxygen consumption (VO_2_peak) or the ventilatory threshold (VO_2_AT) [13], biomarkers [14], hand grip strength [15], and the use of neuroimaging techniques [16]. Although some of the objective measures have been correlated to disease severity, this points to more severe symptomatology. It is not directly related to measures of symptoms. Further studies correlating symptoms to more objective measures of pathophysiology should be done.

Another important aspect of the symptomatology of ME/CFS is that this may change over time and increase in severity during episodes of post-exertional malaise (PEM) [2,17,18]. Many follow-up studies have been published in which the authors investigated the change in severity over time: from complete recovery to deterioration. For an in-depth analysis, see the study of Ghali et al. [19]. Nevertheless, almost all of the published studies have used questionnaires to assess the change in severity, with the inherent potential of response bias.

It has been hypothesized by Low et al. that symptoms of orthostatic intolerance are caused by cerebral hypoperfusion and/or sympathetic activation, but objective proof of this has not been confirmed [20]. In recent studies, we have demonstrated that extracranial Doppler imaging of blood flow through the internal carotid arteries and the vertebral arteries during head-up tilt table testing provides an objective method of measuring changes in cerebral blood flow. These studies have confirmed that ME/CFS patients experience a statistically and clinically significant reduction in cerebral blood flow (CBF) during upright posture, with a mean reduction compared to supine values of 26%, versus just 7% in healthy adult controls [21]. This measure of CBF provides an objective assessment of one of the important physiologic abnormalities in ME/CFS. The hypothesis of the present study is that there is a relationship between the severity of the disease and the degree of CBF reduction during tilt testing.

## 2. Materials and Methods

### 2.1. Participants

To demonstrate this relationship, we used two different approaches: first, we searched our database for ME/CFS patients who underwent two assessments of their ME/CFS symptomatology and who had a tilt tests on the same day as the symptom assessments, one at the first visit and the second during follow-up. The change in symptomatology was related to the change in CBF during the tilt test. Second, we combined the data from two previously published studies of hypermobile patients (n = 200) and of long-haul COVID patients (n = 19) [22,23], where disease severity, as defined by the 2011 international consensus criteria (ICC) [1], was available but not analyzed in the published studies. Eligible patients were drawn from the database of individuals who had undergone head-up tilt testing at the Stichting CardioZorg, an outpatient cardiology clinic specializing in the care of those with ME/CFS. We included individuals who visited the clinic between October 2012 to March 2023 for the assessment of the clinical symptoms of ME/CFS and who underwent a head-up tilt test for the diagnosis of orthostatic intolerance. In this study we selected patients who had returned to our clinic for a re-assessment of symptoms and who underwent a second tilt test after an interval of more than 1 year. The second tilt test was performed for two reasons: 1. to confirm the observed abnormalities of the first tilt test (mainly for social security reasons); in these patients there had been no change in the severity of their complaints, 2. due to worsening of their ME/CFS symptoms. During the first visit, we determined whether patients satisfied the criteria for CFS [24] or ME [1], taking the exclusion criteria into account. No other illnesses provided a sufficient alternative explanation for the symptomatology. Patients were selected for analysis if both a detailed history was documented, including the ICC severity assessment, and adequate CBF data were available in both supine and tilted positions. Patients were excluded from the analysis if they were being treated with drugs influencing HR or BP at the time of the tilt testing or were using sympathomimetic agents to treat lung disease. Where this information was known before the visit, patients were advised to taper and stop those drugs.

The study was carried out in accordance with the Declaration of Helsinki. All ME/CFS participants and healthy controls gave informed, written consent. The study was approved by the medical ethics committee of the Slotervaart Hospital, Amsterdam with protocol-number for ME/CFS patients P1736.

### 2.2. International Consensus Criteria (ICC) Severity of Disease

To classify ME/CFS severity during the first and second visit, we used the ICC criteria. Mild severity required an approximate 50% reduction in pre-illness activity level. Moderate severity required patients to be mostly housebound. Severe patients were mostly bedridden and very severe patients were totally bedridden and needed help with basic functions [1]. Very severe patients were excluded because they were unable to tolerate tilt testing. We have confirmed that the ICC severity classification is valid and correlates with objective measures of physical function, including number of steps per day and exercise capacity as obtained by cardiopulmonary exercise testing [12].

### 2.3. Tilt Test Protocol

Measurements were performed as described previously [21,25]. Briefly, all participants were positioned supine for 20 min before being tilted head-up to 70 degrees. In earlier studies—especially in patients who had not reported orthostatic intolerance symptoms—the duration of the tilt was longer (maximally 30 min). In these studies of a longer tilting duration, a second set of image acquisitions was acquired halfway through the test: mid-tilt acquisition (starting at 13 (3) min post-onset of tilt and ending at 17 (4) min post-tilting), and a third image acquisition was obtained at ≥25 min of the upright position. All second-tilt tests were of shorter duration. Therefore, in patients with a long tilt duration, the mid-tilt data were taken for comparison purposes. The tilt duration was determined by patient wellbeing and symptomatology; when patients experienced increasing severity of symptoms, they were returned to the horizontal position either at their request or by our judgement to avoid syncope. As the aim of the tilt test was quantification of orthostatic intolerance, avoidance of syncope did not interfere with that goal but led to a shorter tilt duration. HR, systolic, and diastolic blood pressures (SBP, DBP) were continuously recorded by finger plethysmography [26,27]. After the test, HR, SBP, and DBP were extracted from the device and imported into an Excel spreadsheet.

The changes in HR and BP were classified as a normal HR-BP response, postural orthostatic tachycardia syndrome (POTS), or orthostatic hypotension (OH) in accordance with the definitions of POTS and OH in consensus statements [28,29,30].

### 2.4. Extracranial Doppler: Cerebral Blood Flow Measurements

Measurements were performed as described previously [21,25]. Internal carotid artery and vertebral artery Doppler flow velocity frames were acquired by one operator (FCV), using a Vivid-I system (GE Healthcare, Hoevelaken, The Netherlands) equipped with a 6–13 MHz linear transducer. Frames were recorded in the supine position approximately 8 min before the onset of the tilt period and while upright once or twice (see tilt protocol). Blood flows of the internal carotid and vertebral arteries were calculated offline by one investigator (CMCvC). Blood flow in each vessel was calculated from the mean blood flow velocities times the vessel cross-sectional area and expressed in mL/minute. Flow in the individual arteries was calculated in 3–6 cardiac cycles and data were averaged. Total CBF was calculated by adding the flow of the four arteries.

### 2.5. Statistical Analysis

Data were analyzed using the statistical package of Graphpad Prism version 9.5.1 (Graphpad software, La Jolla, CA, USA). All continuous data were tested for normal distribution using the D’Agostino and Pearson omnibus normality test and presented as mean and standard deviation (SD) or as median with the interquartile range (IQR) where appropriate. Nominal data were compared using the Chi-square test. Baseline characteristics of groups were compared using Students *t*-test for paired or unpaired data. Also, an ANOVA was performed on the three groups of mild, moderate, and severe disease, with a post hoc Tukey test. Moreover, a two-way ANOVA with interaction analysis was performed where appropriate. To determine the diagnostic accuracy of the change in the ICC criteria (second visit minus first visit) to predict the presence or absence of worsening, we performed an ROC analysis using Prism. The highest sum of the sensitivity and specificity was considered the optimal cut-off point. The same ROC analysis was used to assess the diagnostic value of change in %CBF reduction (visit 2 minus visit 1) for the prediction of unchanged vs. worsening symptoms. Due to the number of comparisons, we choose a conservative *p*-value of <0.01 to be statistically significant.

## 3. Results

Of the patients evaluated with a head-up tilt test in the Stichting CardioZorg between 2012 and 2023, 110 patients had a detailed history on two occasions, as well as two tilt tests conducted on the same day as the symptoms were elicited. None were using medications that affected HR and BP, and none were using compression stockings at the time of testing. All had sufficient data for analysis. As illustrated in Figure 1, of the 110, 39 had a repeat tilt test to document objective abnormalities, but the symptomatology remained unchanged in essence. The other 71 patients were re-tested because of worsening symptoms. The mean interval between the two tests was 2.5 (1.3) years. After taking the mid-tilt data in cases where there was a long first tilt duration (see the correction as explained in the tilt test protocol), the tilt duration of the first test was 10.7 (6.2) minutes and of the second test 11.8 (7.6 min): *p* = 0.68.

Table 1 shows the baseline characteristics during the first test of the group retested without an important change in their clinical condition (n = 39) and of the group retested due to worsening of symptoms (n = 71). No differences in baseline characteristics were found between the groups. Table 2 shows the severity classification according to the International Consensus Criteria (ICC) [1] at visit 1 and 2. During the first visit, the classification of patients who during the second visit were designated as having unchanged symptoms had more severe disease compared to the patients with worsening symptoms (*p* < 0.004). During the second visit, this pattern was reversed: patients with worsening symptoms had more severe disease and compared to the patients with unchanged symptomatology (*p* < 0.0001). Figure 2 is the graphic representation of these results. A ROC analysis showed that an increase of 1 point in severity had a sensitivity of 68% and a specificity of 85% to detect a worsening symptomatology with an area under the curve of 0.77, *p* < 0.0001 (Figure 3).

Table 3 and Table 4 show the hemodynamic results of the first and second tilt tests for the group with unchanged symptoms versus the one with worsening symptoms. The results of the first tilt test showed no differences between the two groups in HR, blood pressures, CBF, or the distribution of patients having a normal HR and blood pressure response, orthostatic hypotension (OH) or postural orthostatic tachycardia syndrome (POTS). The percentage reduction CBF was different but did not reach our conservative statistical significance of <0.01. Comparison of the two groups at the second tilt test showed no differences in HR, blood pressure, or CBF at baseline and at end-tilt. The percent reduction in CBF was significantly larger in the group with worsening symptoms (*p* < 0.0001). Figure 4 shows the percent reduction in CBF for both groups for both tilt tests, and Figure 5 shows the individual CBF data for both groups and tests. A ROC analysis showed that a further decrease in % CBF of −6% or more at the second tilt test was diagnostic for worsening of symptoms with a sensitivity of 82% and specificity of 97% both, with an area under the curve of 0.96, *p* < 0.0001 (Figure 6).

Finally, we combined the data of two previously published studies, where the ICC criteria were present but not mentioned in the reports [21,22]. Figure 7 shows that with increasing disease severity, the %CBF reduction was significantly larger: the %CBF reduction was significantly different between the three groups of mild, moderate and severe disease with *p* values ranging between *p* = 0.0073 and *p* < 0.0001.

## 4. Discussion

We previously reported a significant CBF reduction, as assessed by extracranial Doppler in a large group of ME/CFS patients during a 70 degree tilt test [21], during a 20 degree tilt test [31], and while sitting [32]. Our data are in line with the transcranial Doppler data of Medow et al. in ME/CFS [33] and in other orthostatic intolerance syndromes [34,35]. Moreover, we related the CBF reduction with various symptoms: pain thresholds decreased directly after a tilt test [36], memory and concentration were reduced post-tilt [37], post-exertional malaise symptoms were temporarily increased after a tilt test [38], and orthostatic intolerance symptoms increased with worsening of the CBF abnormalities [21]. The decline in memory and concentration was previously reported by Medow et al. [33]. Importantly, the authors also showed that administration of phenylephrine reduced the CBF abnormalities and the errors on the N-back test [33]. We also demonstrated that the use of compression stockings both improved clinical symptoms and reduced CBF abnormalities [39,40]. A similar design was used by Streeten et al., using military anti-shock trousers to apply a positive external pressure of 45 mmHg during active standing. The inflation of these trousers substantially reduced or abolished the orthostatic intolerance symptoms in ME/CFS patients [41].

In the present study, we tested the hypothesis that more severe ME/CFS symptomatology is related to a larger decrease in CBF during a tilt test. For this purpose, we used two different approaches. One approach used ICC severity [1] data of two previously published tilt table reports, where severity data were available but not analyzed and published [22,23]. The use of the ICC severity score has been validated against the Rand-36 questionnaire, against the maximal oxygen consumption during a bicycle stress test and against a step tracker [42]. The combined data in the present study showed that as the severity of the disease increased, the CBF reduction during the tilt was larger, increasing from −25% in mild patients, −29% in moderately affected patients, to −33% in severe ME/CFS patients. The differences between the three groups were highly significant.

For the other approach we selected patients in our database, who were re-studied after a mean follow-up period of 2.5 years. These patients could be divided into two groups: the first were patients who had a stable symptomatology and were reassessed after the follow-up period to confirm the previously identified abnormalities. This group of patients was restudied mainly for social security reasons. The second group of patients consisted of patients who were restudied because of deterioration in their clinical status. The difference in the two groups, reporting a stable situation or an increased symptomatology, was also reflected in the distribution of mild, moderate, and severe disease according to the ICC criteria: stable patients had a similar distribution in mild, moderate, and severe disease at the first and second visit, while patients with worsening symptoms showed significantly more moderate and severe disease at the second tilt test. Most importantly, those patients with worsening symptoms showed a significantly larger CBF reduction during repeat tilt testing, while patients with a stable symptomatology did not show a further CBF reduction. Both observations suggest that there is a strong relationship between symptom severity and CBF reduction. Thus, not only an acute reduction in CBF by a tilt test (with or without a therapeutic intervention) resulted in a temporarily increase or decrease in symptoms, but also in a chronic situation worsening of symptoms was correlated with a further CBF reduction and a stable clinical condition was associated with unchanged CBF abnormalities. All these studies show that there is a significant relation between ME/CFS symptoms and the abnormal CBF reduction during a tilt test. Given this relation there are two possibilities: symptoms are the cause of a CBF reduction or the CBF reduction is the cause of the symptom increase. The latter is the most likely sequence, that an acute or a repetitive CBF reduction causes or worsens symptoms. The mechanism of this sequence is uncertain. Other mechanisms currently unknown are possible.

An interesting mechanism to consider is that oxygen supply by means of the cerebral flow is insufficient to meet the metabolic demands of the brain, not only during an orthostatic stress but possibly also at rest. When oxygen supply is insufficient for glucose breakdown, lactate may be produced. Increased lactate levels in cerebrospinal fluid have been demonstrated in a variety of studies in ME/CFS patients [43,44]. Moreover, there are similarities in symptoms reported by patients with ME/FS and patients with lactic acidosis [45]. On the other hand, studies have shown that oxygen extraction of the brain increases to compensate for the reduction in CBF in patients with hypotension, and (pre)-syncope [46,47,48]. The increase in oxygen extraction may be up to 80%, and therefore be sufficient to compensate for the CBF reduction of maximally 51% in the present study (data not shown). Nevertheless, in a previous study we showed that the vast large majority of ME/CFS patients developed symptoms of orthostatic intolerance during the tilt test and that the total number of symptoms was positively related to the extent of the %CBF reduction: a larger CBF reduction induced more orthostatic intolerance symptoms during the tilt [24]. This suggests that the oxygen extraction reserve is limited in ME/CFS patients. This needs further investigation. Another possible mechanism to consider is the presence of post-ischemic (repetitively) stunned or hibernating tissue, which has been extensively studied in the myocardium [49]. There are three possible scenarios following an episode of ischemia induced by a CBF reduction. As stated above, ME/CFS patients may show a limited oxygen extraction reserve, resulting in ischemia when CBF decreases. The first scenario is that the CBF reduction/cerebral ischemia is that extensive that necrosis follows. This is highly unlikely as no MRI studies in ME/CFS have found signs of focal necrosis. The second scenario is that ischemia leads to stunning, which results in dysfunction, but recovers over time when the ischemic episode has disappeared. In myocardium this functional recovery may take days to weeks, but in cerebral tissue the time to recovery is unknown. Our data on the slow recovery of CBF are consistent with this mechanism [50]. The third mechanism that can be considered, is that repeated episodes of stunning lead to hibernation, where the tissue is still viable, but dysfunctional, with dedifferentiated cells and a reduced flow. The reduced CBF in ME/CFS has been demonstrated by Biswal et al. [51]. It can be hypothesized that the chronic complaints in ME/CFS, present without a stressor, are related to this hibernation. These are far-reaching hypotheses, but worth exploring.

### 4.1. Strengths

The strengths of this study include the relatively large sample size of ME/CFS patients undergoing repeat tilt testing, as well as the fact that subjects with a stable condition or with a worsening clinical condition were included. Our data show that the subjective feeling of deterioration or being in a stable clinical condition is supported by the objective measure of the CBF: the presence or absence of a change in symptomatology can be reliably detected by the %change in CBF, where a further reduction of at least 6% has a sensitivity of 82% and a specificity of 97%. These data suggest that CBF measurements can potentially be used to monitor disease progression. For this purpose, a prospective study is warranted. Also, data from the ICC method of classifying disease severity suggest that there is a relation between severity and %CBF reduction, but the change in ICC severity with a one-point increase is less predictive for worsening than the %CBF change, with a sensitivity of 68% and a specificity of 85%.

### 4.2. Limitations

The present study comprises a subset of patients who were restudied for a clinical or legal reason: having worsening symptoms or to confirm the previously identified abnormalities during the tilt test. This may have led to a selection bias. Furthermore, in earlier studies, a longer tilt test duration of around 25 min was used [21], while in later studies, we reduced the tilt duration to maximally 15 min in an attempt to reduce the post-exertional malaise provoked by the test. To overcome the differences in tilt duration, the mid-tilt data of those with a tilt test of long duration were used to compare with the end-tilt data of patients with a shorter tilt duration. In the aforementioned study [21], we demonstrated that the largest change in CBF reduction was present at mid-tilt compared to the supine measurements, but there was also a further CBF reduction from mid-tilt to end-tilt. This previous study shows that with being upright longer, the abnormality in the %CBF difference grows larger. From clinical follow-up, patients report a longer duration of PEM when tilted longer, which is a reason to choose a shorter duration. To analyze comparable data, we used the mid-tilt data from the longer tilt test with the end-tilt data from the shorter tilt tests. The magnitude of changes in longer tilt durations remains speculative, and we are confident that it would not grossly change the results.

In this study, we combined the data of patients with a normal HR-BP response with POTS and OH. We did not have a large enough sample of patients to compare whether individuals in each hemodynamic group with worsening symptoms or with a stable condition had different CBF responses.

The relationship between the change in symptom severity and %CBF change could be strengthened by the inclusion of patients who had reported symptomatic improvement. This needs to be explored in a prospective study. Also, we assessed symptom severity by taking a history, which may be a subjective interpretation by the treating physician. We previously demonstrated that the ICC severity scales correlated with the number of steps per day, as assessed with an activity tracker, and with the %oxygen consumption at the ventilatory threshold and at peak exercise [42], where an increased severity resulted in lower values of the three parameters. These three validated parameters have also been used in a small subgroup of patients with worsening symptoms, showing that the number of steps and %oxygen consumption were reduced after symptom worsening [12]. Thus, although the ICC severity grading may be subject to personal bias, its validity is established.

## 5. Conclusions

In the present study, we showed that the %CBF reduction during tilt testing is stable in patients with unaltered symptoms after a mean of 2.5 years, while in patients whose symptoms worsen, there is a decline in %CBF reduction during the repeat tilt test. These observations are supported by the findings of a larger %CBF reduction in more severe patients as assessed by the ICC criteria. The objectively assessed further reduction in %CBF has a high sensitivity and specificity to confirm worsening of symptoms and has the potential to guide treatment, measure the efficacy of therapy in clinical trials, and help with the assessment of disability status.

## Figures and Tables

**Figure 1 medicina-59-02153-f001:**
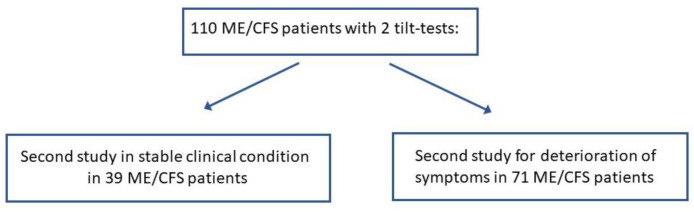
Flow diagram of study population.

**Figure 2 medicina-59-02153-f002:**
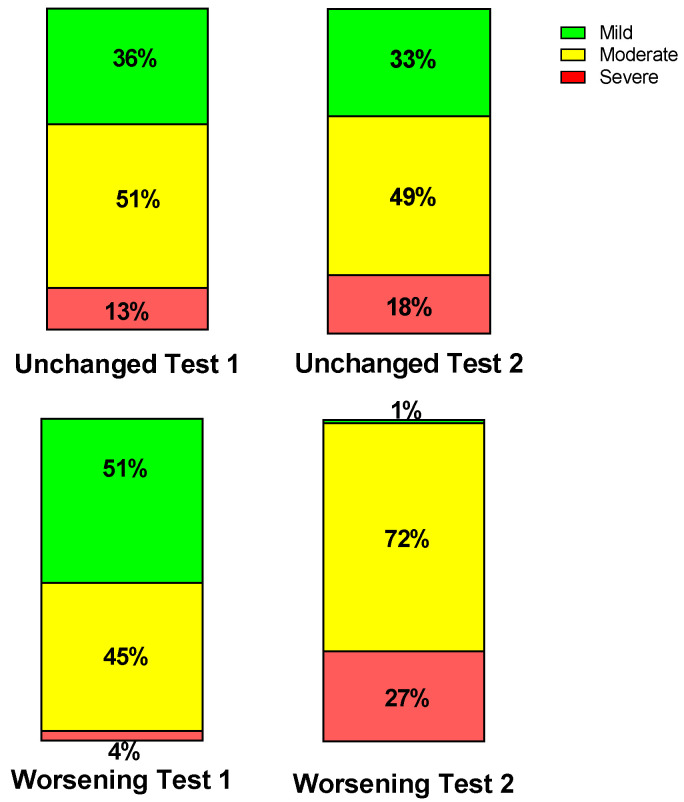
Percent severity per group per test according to International Consensus Criteria presented per test per group. The two stacked upper columns show the unchanged group and the two stacked lower columns the worsening group. The results of test one are on the left, and the right the results of test two are on the right.

**Figure 3 medicina-59-02153-f003:**
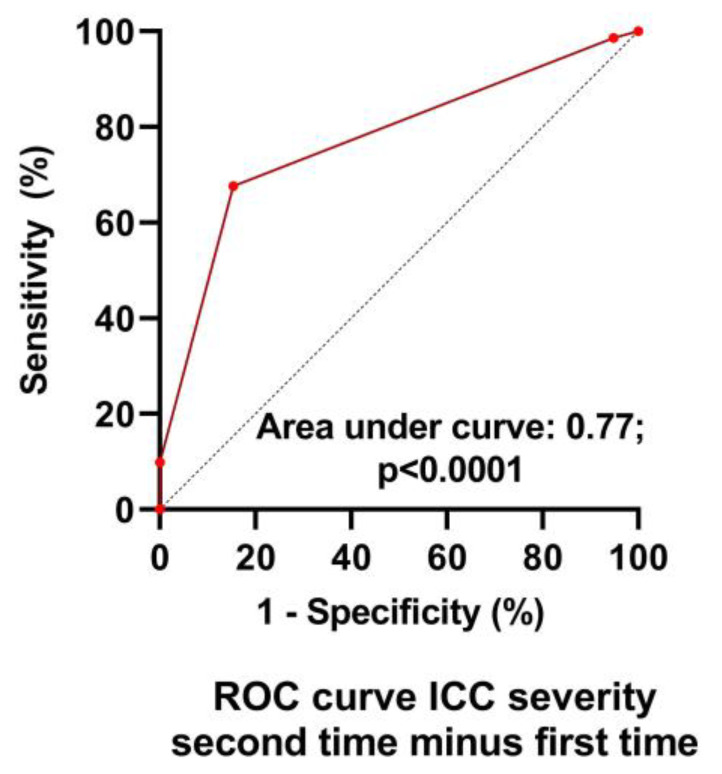
ROC analysis of the change in ICC severity scale to detect worsening symptoms. ICC: international consensus criteria of disease severity [1].

**Figure 4 medicina-59-02153-f004:**
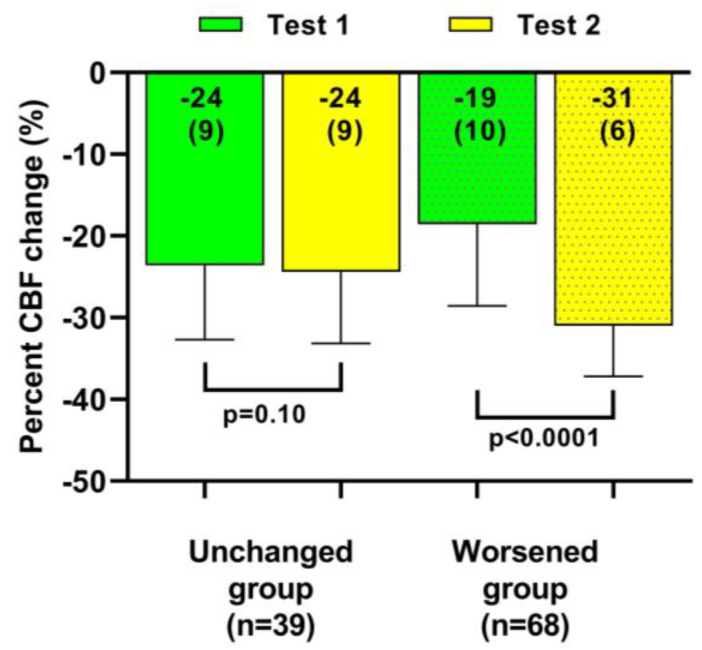
Percent cerebral blood flow change in both head-up tilt tests, comparing patients with a relatively stable clinical condition to those with a worsening condition. CBF: cerebral blood flow.

**Figure 5 medicina-59-02153-f005:**
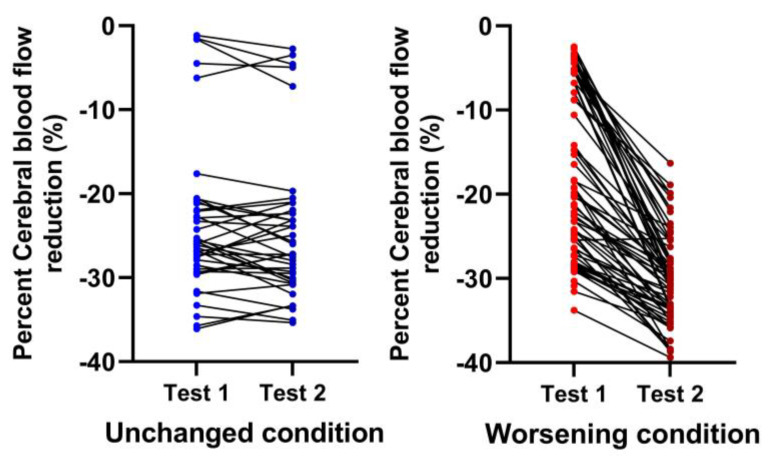
Percent cerebral blood flow individual changes in both head-up tilt tests comparing patients with a relatively stable clinical condition (blue) to those with a worsening condition (red). Unchanged condition meaning patients returning to patient care without a significant change in their clinical situation (blue; left sided panel) compared to worsening condition where patients returned for re-evaluation because of worsening symptomatology (red; right sided panel).

**Figure 6 medicina-59-02153-f006:**
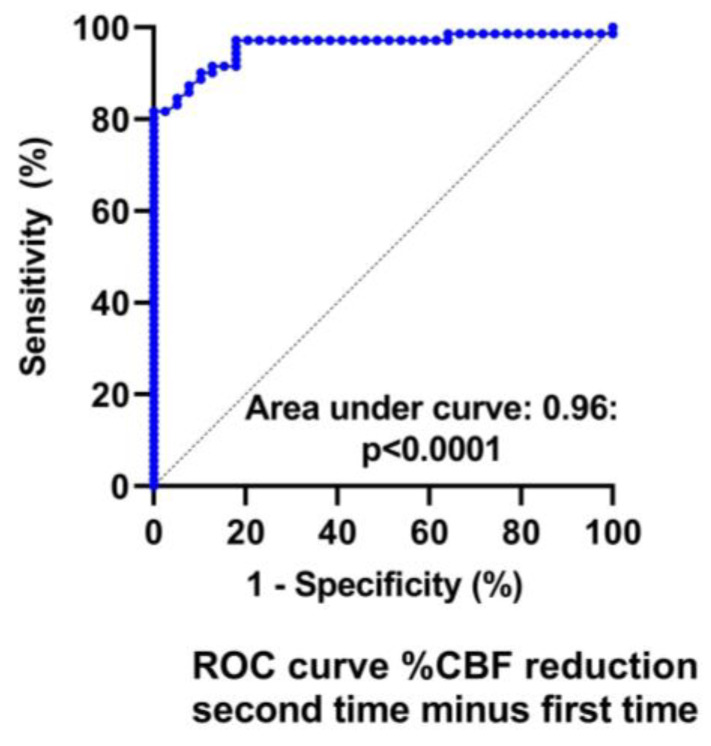
ROC analysis of the change in %CBF reduction to detect worsening symptoms. %CBF reduction: reduction in the percent cerebral blood flow during tilt testing.

**Figure 7 medicina-59-02153-f007:**
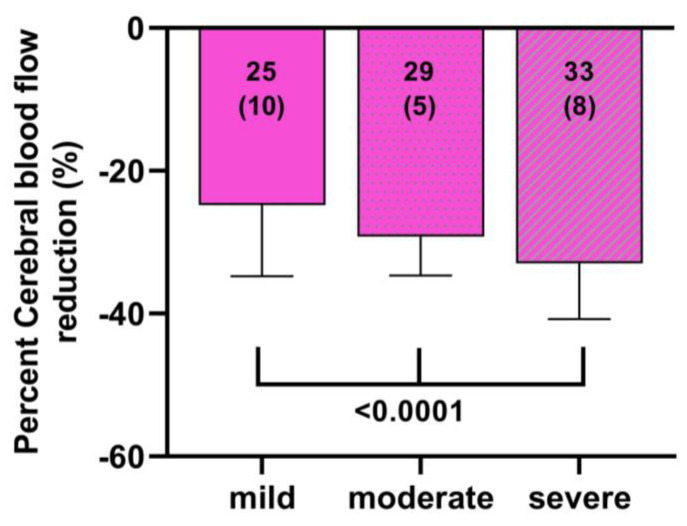
Percent cerebral blood flow reduction in a ME/CFS patient group derived from two previous manuscripts divided by ICC disease severity [21,22]. Post hoc results F (2, 226) = 16.14 *p* < 0.0001; mild versus moderate *p* = 0.0007, mild versus severe *p* < 0.0001 and moderate versus severe *p* = 0.0073.

**Table 1 medicina-59-02153-t001:** Baseline characteristics of the ME/CFS patients comparing patients with a repeat test without change in clinical condition to patients with worsening symptoms.

	No Change (n = 39)	Worsening (n = 71)	*p*-Value
Male/female *	6/33	11/60	0.82
Age (years)	42 (13)	39 (12)	0.13
Height (cm)	171 (7)	172 (8)	0.35
Weight (kg) #	70 (59–85)	73 (62–85)	0.29
BMI (kg/m^2^) #	24.2 (20.3–27.7)	24.0 (21–29.4)	0.46
BSA (m^2^)	1.83 (0.20)	1.87 (0.20)	0.22
Disease duration (years) #	8 (3–13)	11 (6–21)	0.054

* Chi-square 2 × 2 or 2 × 3 analysis; # Median (IQR) Mann–Whitney U test; BMI: body mass index: BSA: body surface area (formula duBois); ME/CFS: myalgic encephalomyelitis/chronic fatigue syndrome; A *p*-value of <0.01 is considered statistically significant.

**Table 2 medicina-59-02153-t002:** Severity of disease change of the ME/CFS patients comparing patients with a repeat test without change in clinical condition to patients with worsening symptoms.

	No Change (n = 39)	Worsening (n = 71)	*p*-Value
Test 1: mild/moderate/severe *	14/20/5 (36/51/13%)	36/32/3 (51/45/4%)	0.004
Test 2: mild/moderate/severe *	13/19/7 (33/49/18%)	1/51/19 (1/72/27%)	<0.0001

* Chi-square analysis (3 × 2 table); ME/CFS: myalgic encephalomyelitis/chronic fatigue syndrome.

**Table 3 medicina-59-02153-t003:** Hemodynamic results of the ME/CFS patients comparing patients with a repeat tilt test without change in clinical condition with patients with worsening symptoms: tilt test one.

	No Change (n = 39)	Worsening (n = 71)	*p*-Value
Hemodynamic result tilt test (normHRBP/OH/POTS)	26/7/6 (67/18/15%)	47/4/17 (69/6/25%)	0.10
HR supine (bpm)	70 (11)	71 (11)	0.78
HR end-tilt (bpm)	88 (17)	92 (16)	0.28
SBP supine (mmHg)	139 (22)	132 (23)	0.11
SBP end-tilt (mmHg)	132 (22)	131 (18)	0.73
DBP supine (mmHg)	81 (11)	78 (12)	0.18
DBP end-tilt (mmHg)	86 (14)	85 (9)	0.64
CBF supine (mL/min)	608 (90)	605 (103)	0.88
CBF end-tilt (mL/min)	463 (78)	494 (108)	0.12
%CBF reduction (%)	24 (9)	19 (10)	0.01

HR: heart rate: SBP: systolic blood pressure; DBP: diastolic blood pressure; ME/CFS: myalgic encephalomyelitis/chronic fatigue syndrome; CBF: cerebral blood flow.

**Table 4 medicina-59-02153-t004:** Hemodynamic results of the ME/CFS patients comparing patients with a repeat tilt test without change in clinical condition with patients with worsening symptoms: tilt test two.

	No Change (n = 39)	Worsening (n = 71)	*p*-Value
Hemodynamic result tilt test (normHRBP/OH/POTS)	28/4/7 (72/10/18%)	41/8/22 (59/10/31%)	0.30
HR supine (bpm)	70 (8)	70 (11)	0.74
HR end-tilt (bpm)	87 (17)	92 (17)	0.17
SBP supine (mmHg)	136 (21)	134 (17)	0.58
SBP end-tilt (mmHg)	128 (25)	126 (18)	0.70
DBP supine (mmHg)	81 (11)	78 (12)	0.22
DBP end-tilt (mmHg)	86 (14)	85 (10)	0.74
CBF supine (ml/min)	586 (88)	616 (103)	0.13
CBF end-tilt (ml/min)	441 (73)	426 (88)	0.36
%CBF reduction (%)	24 (9)	31 (6)	<0.0001

HR: heart rate: SBP: systolic blood pressure; DBP: diastolic blood pressure; ME/CFS: myalgic encephalomyelitis/chronic fatigue syndrome; CBF: cerebral blood flow.

## Data Availability

The datasets analyzed in the current study are available from the corresponding author on reasonable request.

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
