# Peer review of "Worsening Symptoms Is Associated with Larger Cerebral Blood Flow Abnormalities during Tilt-Testing in Myalgic Encephalomyelitis/Chronic Fatigue Syndrome (ME/CFS)"

_medicina, 2023, doi:10.3390/medicina59122153_

Round 1

Reviewer 1 Report

Comments and Suggestions for Authors

This is a manuscript correlating severity of ME/CFS symptoms to reduction in cerebral blood flow as induced by tilt testing.  As such it is an important work because it demonstrates that our understanding of physiology and pathophysiology as applied to understanding the cause of symptom complaints is a valid approach to investigate the cause of this disease's symptoms.

The manuscript is well written except for some concerns expressed below.

My major concern is something not addressed in the manuscript and perhaps should not be addressed in the manuscript.  Nevertheless, I feel compelled to raise the issue and leave it to the authors and editors to decide if any alteration of the manuscript is appropriate:  The ICC criteria used to characterize ME/CFS patients is not the current criteria used for diagnosis - at least in the U.S.  I understand why the authors chose to use the ICC criteria but wonder if the manuscript's silence on not using other criteria, or addressing how the use of this criteria relates to the more current (or perhaps popular) diagnostic criteria is the elephant in the room?

Otherwise, except for the few concerns below, the manuscript is ready to be printed (in my opinion).

Lines 43-48:

Therefore, these questionnaires should preferably be used in conjunction with objective measures of symptoms. Examples of these objective measures are the use of heart rate (HR) changes [8-10], of heart rate variability (HRV) differences [9, 11], the use of activity trackers [12], and peak oxygen consumption (VO2peak) or at the ventilatory threshold (VO2AT) [13], of biomarkers [14], of hand grip strength [15], and the use of neuroimaging techniques [16].

The examples given are not proven objective measures of symptoms.  Our knowledge of physiology and pathophysiology suggest that there might be correlations between these objective measures and the symptoms exhibited but these have not been definitively proven.  Accordingly, kindly reword.

Lines 56-58:

It has been hypothesized by Low et al that symptoms of orthostatic intolerance are 56 caused by cerebral hypoperfusion and/or sympathetic activation, but objective proof of this has not been confirmed [20].

If Low et al. did not provide objective proof, move reference to more appropriate place.

It has been hypothesized by Low et al. [20] that symptoms of orthostatic intolerance are 56 caused by cerebral hypoperfusion and/or sympathetic activation, but objective proof of this has not been confirmed.

Lines 68-71:

 we searched our database for ME/CFS patients who underwent two assessments of their ME/CFS symptomatology and who had two tilt tests on the same day as the symptom assessments, one at the first visit and the second during follow-up.

A tilt test at the first visit and a tilt test on the follow-up visit are not tilt tests on the same day!

Lines 112-114:

In earlier studies –especially patients had not reported orthostatic intolerance symptoms-the duration of tilt was longer (maximally 30 minutes)

Insert who

In earlier studies –especially patients who had not reported orthostatic intolerance symptoms-the duration of tilt was longer (maximally 30 minutes)

Lines 136-137:

Blood flow of the internal carotid and vertebral arteries was calculated offline by one investigator (CMCvC).

Blood flows of arteries were calculated.

Lines 285-286:

Both observations suggest that there is a strong relationship between symptom severity and CBF reduction

I think the authors meant to say that there is a strong relationship between symptom severity and CBF reduction during tilt testing.  I don’t think the authors present sufficient data to state that there is a reduction in CBF in the patients throughout their day as they attempt daily living.

Line 296:

An interesting mechanism to consider is that oxygen supply by means of the cerebral.....

This paragraph presents three possible mechanisms known to the authors.  The tone at the end of the paragraph is that these are the only three mechanisms possible.  The reviewer disagrees.  Other mechanisms currently unknown are possible.  Paragraph needs rewording.

Lines 306-307:

Nevertheless, in a previous study we showed that the vast large majority of ME/CFS developed symptoms of orthostatic intolerance during the tilt test

Nevertheless, in a previous study we showed that the vast large majority of ME/CFS patients developed symptoms of orthostatic intolerance during the tilt test

Lines 316-317:

The first scenario is that the CBF reduction/cerebral ischemia is that extensive that necrosis follows

Reviewer is uncertain as to what this sentence means.  Do the authors mean to say that the first scenario is that the CBF reduction/cerebral ischemia is extensive and that necrosis follows?

Lines 352-355:

Therefore, it is conceivable that if all patients were exposed to a longer tilt duration, the %CBF differences between the first and second tilt might have differed from the data in the present study where short tilt durations were compared, but we think this is unlikely.

If authors are going to state their opinion that they think the difference is exposure to tilt was unlikely to affect the data, they are obligated to explain why.

Lines 43-48:

Therefore, these questionnaires should preferably be used in conjunction with objective measures of symptoms. Examples of these objective measures are the use of heart rate (HR) changes [8-10], of heart rate variability (HRV) differences [9, 11], the use of activity trackers [12], and peak oxygen consumption (VO2peak) or at the ventilatory threshold (VO2AT) [13], of biomarkers [14], of hand grip strength [15], and the use of neuroimaging techniques [16].

 The examples given are not proven objective measures of symptoms.  Our knowledge of physiology and pathophysiology suggest that there might be correlations between these objective measures and the symptoms exhibited but these have not been definitively proven.  Accordingly, kindly reword.

Lines 56-58:

It has been hypothesized by Low et al. that symptoms of orthostatic intolerance are caused by cerebral hypoperfusion and/or sympathetic activation, but objective proof of this has not been confirmed [20].

If Low et al. did not provide objective proof, move reference to more appropriate place.

It has been hypothesized by Low et al. [20] that symptoms of orthostatic intolerance are 56 caused by cerebral hypoperfusion and/or sympathetic activation, but objective proof of this has not been confirmed.

Lines 68-71:

 we searched our database for ME/CFS patients who underwent two assessments of their ME/CFS symptomatology and who had two tilt tests on the same day as the symptom assessments, one at the first visit and the second during follow-up.

A tilt test at the first visit and a tilt test on the follow-up visit are not tilt tests on the same day!  Please clarify what is meant.

Lines 112-114:

In earlier studies –especially patients had not reported orthostatic intolerance symptoms-the duration of tilt was longer (maximally 30 minutes)

Insert who

In earlier studies –especially patients who had not reported orthostatic intolerance symptoms-the duration of tilt was longer (maximally 30 minutes)

Lines 136-137:

Blood flow of the internal carotid and vertebral arteries was calculated offline by one investigator (CMCvC).

Blood flows of arteries were calculated.  (Two arteries with two different blood flows.)

Lines 285-286:

Both observations suggest that there is a strong relationship between symptom severity and CBF reduction

I think the authors meant to say that there is a strong relationship between symptom severity and CBF reduction during tilt testing.  I don’t think the authors present sufficient data to state that there is a reduction in CBF in the patients throughout their day as they attempt daily living.

Line 296:

An interesting mechanism to consider is that oxygen supply by means of the cerebral....

This paragraph presents three possible mechanisms known to the authors.  The tone at the end of the paragraph is that these are the only three mechanisms possible.  The reviewer disagrees.  Other mechanisms currently unknown are possible.  Paragraph needs rewording.

Lines 306-307:

Nevertheless, in a previous study we showed that the vast large majority of ME/CFS developed symptoms of orthostatic intolerance during the tilt test

Nevertheless, in a previous study we showed that the vast large majority of ME/CFS patients developed symptoms of orthostatic intolerance during the tilt test

Lines 316-317:

The first scenario is that the CBF reduction/cerebral ischemia is that extensive that necrosis follows.

Reviewer is uncertain as to what this sentence means.  Do the authors mean to say that the first scenario is that the CBF reduction/cerebral ischemia is extensive and that necrosis follows?

Lines 352-355:

Therefore, it is conceivable that if all patients were exposed to a longer tilt duration, the %CBF differences between the first and second tilt might have differed from the data in the present study where short tilt durations were compared, but we think this is unlikely.

If authors are going to state their opinion that they think the difference in duration to exposure to tilt was unlikely to affect the data, they are obligated to explain why.

Comments on the Quality of English Language

Minor clarifications would be appreciated as indicated in the review above.

Author Response

Open Review

Quality of English Language

( ) I am not qualified to assess the quality of English in this paper
( ) English very difficult to understand/incomprehensible
( ) Extensive editing of English language required
( ) Moderate editing of English language required
(x) Minor editing of English language required
( ) English language fine. No issues detected

Yes

Can be improved

Must be improved

Not applicable

Does the introduction provide sufficient background and include all relevant references?

(x)

( )

( )

( )

Are all the cited references relevant to the research?

(x)

( )

( )

( )

Is the research design appropriate?

(x)

( )

( )

( )

Are the methods adequately described?

( )

(x)

( )

( )

Are the results clearly presented?

( )

( )

( )

( )

Are the conclusions supported by the results?

( )

( )

( )

( )

Comments and Suggestions for Authors

This is a manuscript correlating severity of ME/CFS symptoms to reduction in cerebral blood flow as induced by tilt testing.  As such it is an important work because it demonstrates that our understanding of physiology and pathophysiology as applied to understanding the cause of symptom complaints is a valid approach to investigate the cause of this disease's symptoms.

 We kindly thank the reviewer for his thorough comments to improve the manuscript.

The manuscript is well written except for some concerns expressed below.

My major concern is something not addressed in the manuscript and perhaps should not be addressed in the manuscript.  Nevertheless, I feel compelled to raise the issue and leave it to the authors and editors to decide if any alteration of the manuscript is appropriate:  The ICC criteria used to characterize ME/CFS patients is not the current criteria used for diagnosis - at least in the U.S.  I understand why the authors chose to use the ICC criteria but wonder if the manuscript's silence on not using other criteria, or addressing how the use of this criteria relates to the more current (or perhaps popular) diagnostic criteria is the elephant in the room?

We check patients for Fukuda (especially with PEM included), SEID AND ICC criteria. We use the latter especially because the international consensus paper defines the severity of disease criteria in an effort to make grading in scientific work possible as well as making groups more comparable. Which criteria are used in US by the reviewer we are not familiar with, but as stated we use several to be as thorough as possible.

Otherwise, except for the few concerns below, the manuscript is ready to be printed (in my opinion).

Lines 43-48:

Therefore, these questionnaires should preferably be used in conjunction with objective measures of symptoms. Examples of these objective measures are the use of heart rate (HR) changes [8-10], of heart rate variability (HRV) differences [9, 11], the use of activity trackers [12], and peak oxygen consumption (VO2peak) or at the ventilatory threshold (VO2AT) [13], of biomarkers [14], of hand grip strength [15], and the use of neuroimaging techniques [16].

The examples given are not proven objective measures of symptoms.  Our knowledge of physiology and pathophysiology suggest that there might be correlations between these objective measures and the symptoms exhibited but these have not been definitively proven.  Accordingly, kindly reword.

 Although maybe disease severity says something about symptom severity, the statement is correct. Additional comments have been made. Further studies correlating symptoms with pathophysiology should be done.

Lines 56-58:

It has been hypothesized by Low et al that symptoms of orthostatic intolerance are 56 caused by cerebral hypoperfusion and/or sympathetic activation, but objective proof of this has not been confirmed [20].

If Low et al. did not provide objective proof, move reference to more appropriate place.

Reference 20 is the reference of Low et al where he states this comment. We added a reference to the mentioned abnormalities in cerebral flow of ME/CFS patients compared to healthy controls.

It has been hypothesized by Low et al. [20] that symptoms of orthostatic intolerance are 56 caused by cerebral hypoperfusion and/or sympathetic activation, but objective proof of this has not been confirmed.

Lines 68-71:

 we searched our database for ME/CFS patients who underwent two assessments of their ME/CFS symptomatology and who had two tilt tests on the same day as the symptom assessments, one at the first visit and the second during follow-up.

A tilt test at the first visit and a tilt test on the follow-up visit are not tilt tests on the same day!

Completely right: two was changes in a, as symptom assessment WAS done on the same day as the tilt-test.

Lines 112-114:

In earlier studies –especially patients had not reported orthostatic intolerance symptoms-the duration of tilt was longer (maximally 30 minutes)

Insert who added

In earlier studies –especially patients who had not reported orthostatic intolerance symptoms-the duration of tilt was longer (maximally 30 minutes)

Lines 136-137:

Blood flow of the internal carotid and vertebral arteries was calculated offline by one investigator (CMCvC).

Blood flows of arteries were calculated. adapted

Lines 285-286:

Both observations suggest that there is a strong relationship between symptom severity and CBF reduction

I think the authors meant to say that there is a strong relationship between symptom severity and CBF reduction during tilt testing.  I don’t think the authors present sufficient data to state that there is a reduction in CBF in the patients throughout their day as they attempt daily living.

We respectfully disagree with the reviewer. Patients were symptom/severity assessed on the day of the tilt and symptomatology/disease severity – as ICC criteria include a broader range of symptoms as IOM criteria has – was based on daily live symptoms. We believe CBF reduction during the test reflects the general condition of the patient. We agree of course that getting more detailed information on what is happening in the brain in daily life would be a very interesting topic of research, hopefully giving a broader insight into what is happening in this disease.

Line 296:

An interesting mechanism to consider is that oxygen supply by means of the cerebral.....

This paragraph presents three possible mechanisms known to the authors.  The tone at the end of the paragraph is that these are the only three mechanisms possible.  The reviewer disagrees.  Other mechanisms currently unknown are possible.  Paragraph needs rewording. We added the suggestion of the reviewer.

Lines 306-307:

Nevertheless, in a previous study we showed that the vast large majority of ME/CFS developed symptoms of orthostatic intolerance during the tilt test

Nevertheless, in a previous study we showed that the vast large majority of ME/CFS patients developed symptoms of orthostatic intolerance during the tilt test added

Lines 316-317:

The first scenario is that the CBF reduction/cerebral ischemia is that extensive that necrosis follows

Reviewer is uncertain as to what this sentence means.  Do the authors mean to say that the first scenario is that the CBF reduction/cerebral ischemia is extensive and that necrosis follows?

It was presented as a possible option, but the following sentence immediately dismissed this option as we haven’t seen any stroke like presentations of ME/CFS patients with permanent damage, which would be present IF necroses did happen.

Lines 352-355:

Therefore, it is conceivable that if all patients were exposed to a longer tilt duration, the %CBF differences between the first and second tilt might have differed from the data in the present study where short tilt durations were compared, but we think this is unlikely.

If authors are going to state their opinion that they think the difference is exposure to tilt was unlikely to affect the data, they are obligated to explain why. We rephrased this sentence to clarify.

Comments on the Quality of English Language

Minor clarifications would be appreciated as indicated in the review above.

Submission Date

19 November 2023

Date of this review

28 Nov 2023 02:17:43

Reviewer 2 Report

Comments and Suggestions for Authors

This manuscript documents a relationship between cerebral blood flow reduction and severity of ME/CFS, which is a useful contribution.  The investigators examined individuals who had repeat testing and could place them into two groups—ones with no change in severity vs those with increased severity, and increased severity was associated with greater abnormality in CBF. 

They also used CBF data from two prior papers that they had published in which they had not considered symptom severity when examining the abnormalities in CBF.  I have some questions concerning this part of the study.  The authors write “We combined the data from two previously published studies of hypermobile patients (n-200) and of long Haul COVID patients (n-19).  When I looked up the long COVID study, it said that 14 long COVID patients were compared to 14 ME/CFS patients.   Is the number 19 incorrect?  Did the authors combine long COVID patient data with ME/CFS patient data?  While symptoms overlap between the two groups, it is currently not warranted to merely lump long COVID with ME/CFS patients who became ill before the pandemic.  If they did include long COVID in the analysis of the prior data, I think that they should display the results from those patients separately to let us know if/how they may differ from the 200 ME/CFS patients. If they instead used the data from the 14 ME/CFS patients mentioned in the long COVID study, then they will just need to correct the number.

Were any of the retested patients ones with long COVID?  None could be long COVID patients in the “worsening” group since the disease duration is 6 to 21 years. But I see that some of the “no change” patients were ill only 3 yrs and thus may or may not have become ill before the pandemic. If some long COVID patients are included this group, this fact should be indicated, and how many.  I think it would be better not to mix the two diagnoses.

One minor revision for lines 44-48—too much use of “of”.   Corrected version is:

Examples of these objective measures are the use of heart rate changes, heart rate variability, differences, activity trackers, peak oxygen consumption or the ventilatory threshold, biomarkers, hand grip strength, or neuroimaging techniques.  (I didn’t add the abbreviations and citation numbers).

The text in the colored boxes and for the key to Fig 2 should be larger.  White lettering would be easier to see when the box color is dark.

Author Response

Open Review

Quality of English Language

( ) I am not qualified to assess the quality of English in this paper
( ) English very difficult to understand/incomprehensible
( ) Extensive editing of English language required
( ) Moderate editing of English language required
( ) Minor editing of English language required
(x) English language fine. No issues detected

Yes

Can be improved

Must be improved

Not applicable

Does the introduction provide sufficient background and include all relevant references?

(x)

( )

( )

( )

Are all the cited references relevant to the research?

(x)

( )

( )

( )

Is the research design appropriate?

(x)

( )

( )

( )

Are the methods adequately described?

( )

(x)

( )

( )

Are the results clearly presented?

( )

(x)

( )

( )

Are the conclusions supported by the results?

(x)

( )

( )

( )

Comments and Suggestions for Authors

This manuscript documents a relationship between cerebral blood flow reduction and severity of ME/CFS, which is a useful contribution.  The investigators examined individuals who had repeat testing and could place them into two groups—ones with no change in severity vs those with increased severity, and increased severity was associated with greater abnormality in CBF. 

They also used CBF data from two prior papers that they had published in which they had not considered symptom severity when examining the abnormalities in CBF.  I have some questions concerning this part of the study.  The authors write “We combined the data from two previously published studies of hypermobile patients (n-200) and of long Haul COVID patients (n-19).  When I looked up the long COVID study, it said that 14 long COVID patients were compared to 14 ME/CFS patients.   Is the number 19 incorrect?  

The number of long covid is indeed correct as it is not the study comparing long covid with ME/CFS EBV and insidious which indeed included 14 patients, but the other study reporting on pots prevalence.

Did the authors combine long COVID patient data with ME/CFS patient data?  While symptoms overlap between the two groups, it is currently not warranted to merely lump long COVID with ME/CFS patients who became ill before the pandemic.  

Considering the same palette of symptoms and the same objective abnormalities of CBF during upright orthostatic stress that long COVID IS ME/CFS due to covid. If EBV and CMV and Lyme for instance can introduce ME/CFS, why could a virus that is capable of creating a pandemic cannot create the same disease. So considering the studies reported long covid IS ME/CFS due to covid and data can therefore be combined in analysis.

If they did include long COVID in the analysis of the prior data, I think that they should display the results from those patients separately to let us know if/how they may differ from the 200 ME/CFS patients. If they instead used the data from the 14 ME/CFS patients mentioned in the long COVID study, then they will just need to correct the number.

The number of 19 would be to little to compare to 200 ME/CFS due to other causes to display. Adding the covid information as this group of patients is expanding fast and creating an huge increase in the ME/CFS outbreaks/epidemic adds to the relationship. Leaving this group out would not change a lot, but we like to emphasize long covid as part of the problem.

Were any of the retested patients ones with long COVID?  None could be long COVID patients in the “worsening” group since the disease duration is 6 to 21 years. But I see that some of the “no change” patients were ill only 3 yrs and thus may or may not have become ill before the pandemic. If some long COVID patients are included this group, this fact should be indicated, and how many.  I think it would be better not to mix the two diagnoses.

Since this article 1 long covid patient has been retested with important worsening and increase in CBF abnormalitiy.

One minor revision for lines 44-48—too much use of “of”.   Corrected version is:

Examples of these objective measures are the use of heart rate changes, heart rate variability, differences, activity trackers, peak oxygen consumption or the ventilatory threshold, biomarkers, hand grip strength, or neuroimaging techniques.  (I didn’t add the abbreviations and citation numbers).

Thanks for this suggestion, we adapted the sentence.

The text in the colored boxes and for the key to Fig 2 should be larger.  White lettering would be easier to see when the box color is dark.

Adapted as suggested, however white lettering will not go well with the bright yellow. Will adapt the shading.

Submission Date

19 November 2023

Date of this review

04 Dec 2023 01:00

Reviewer 3 Report

Comments and Suggestions for Authors

Please see my suggestions/comments to the reviewed manuscript:

Introduction

I suggest that the Introduction section to be finished at the following phrase: “The hypothesis of 65 the present study is that there is a relation between the severity of the disease and the 66 degree of CBF reduction during tilt testing”. The paragraph that follows this phrase would be better positioned under the methods section, as it describes the approach used to recruit the study population.

Material and Methods

In the item “2.1 Participants” the sample size for the study is not clear. I would recommend the authors to state how many participants were initially selected (from the authors’ database) to the study, according to their selection criteria (i.e. those who had assessments and tilt-tests on the same day). Furthermore, I would suggest the authors to state the number of people with ME/CFS in their database, to provide the readers a clearer information on this study’s sample size.

The additional items 2.2 to 2.5 are well described and well referenced.

Results

The study results were well presented with tables and charts that illustrate well the findings from the reported study. My only suggestion to this section is that the authors use the proper statistical notations, such as P=0.00  or P<0.05, instead of p=ns (see page 4, end of the paragraph).

Discussion

The authors considered their findings in the light of the significant literature on ME/CFS pathophysiology, particularly taking into account the potential abnormal mechanisms of larger cerebral blood flow. Their findings could explain some of the symptoms presented in people with ME/CFS, particularly on those who have severe symptoms.

Author Response

Open Review

Quality of English Language

( ) I am not qualified to assess the quality of English in this paper
( ) English very difficult to understand/incomprehensible
( ) Extensive editing of English language required
( ) Moderate editing of English language required
( ) Minor editing of English language required
(x) English language fine. No issues detected

Yes

Can be improved

Must be improved

Not applicable

Does the introduction provide sufficient background and include all relevant references?

( )

(x)

( )

( )

Are all the cited references relevant to the research?

(x)

( )

( )

( )

Is the research design appropriate?

(x)

( )

( )

( )

Are the methods adequately described?

( )

(x)

( )

( )

Are the results clearly presented?

(x)

( )

( )

( )

Are the conclusions supported by the results?

(x)

( )

( )

( )

Comments and Suggestions for Authors

Please see my suggestions/comments to the reviewed manuscript:

Introduction

I suggest that the Introduction section to be finished at the following phrase: “The hypothesis of 65 the present study is that there is a relation between the severity of the disease and the 66 degree of CBF reduction during tilt testing”. The paragraph that follows this phrase would be better positioned under the methods section, as it describes the approach used to recruit the study population.

 On the request of the reviewer, the section was moved to 2.1 method section.

Material and Methods

In the item “2.1 Participants” the sample size for the study is not clear. I would recommend the authors to state how many participants were initially selected (from the authors’ database) to the study, according to their selection criteria (i.e. those who had assessments and tilt-tests on the same day). Furthermore, I would suggest the authors to state the number of people with ME/CFS in their database, to provide the readers a clearer information on this study’s sample size.

The sample size was clarified with a patient flow diagram in figure 1.

The additional items 2.2 to 2.5 are well described and well referenced.

Results

The study results were well presented with tables and charts that illustrate well the findings from the reported study. My only suggestion to this section is that the authors use the proper statistical notations, such as P=0.00  or P<0.05, instead of p=ns (see page 4, end of the paragraph). altered

Discussion

The authors considered their findings in the light of the significant literature on ME/CFS pathophysiology, particularly taking into account the potential abnormal mechanisms of larger cerebral blood flow. Their findings could explain some of the symptoms presented in people with ME/CFS, particularly on those who have severe symptoms.

Submission Date

19 November 2023

Date of this review

05 Dec 2023 22:53:59